# Electrochemotherapy in Vulvar Cancer and Cisplatin Combined with Electroporation. Systematic Review and In Vitro Studies

**DOI:** 10.3390/cancers13091993

**Published:** 2021-04-21

**Authors:** Anna Myriam Perrone, Gloria Ravegnini, Stefano Miglietta, Lisa Argnani, Martina Ferioli, Eugenia De Crescenzo, Marco Tesei, Marco Di Stanislao, Giulia Girolimetti, Giuseppe Gasparre, Anna Maria Porcelli, Francesca De Terlizzi, Claudio Zamagni, Alessio Giuseppe Morganti, Pierandrea De Iaco

**Affiliations:** 1Division of Oncologic Gynecology, IRCCS—Azienda Ospedaliero-Universitaria di Bologna, 40138 Bologna, Italy; myriam.perrone@aosp.bo.it (A.M.P.); eugenia.decrescenzo@studio.unibo.it (E.D.C.); marco.tesei@aosp.bo.it (M.T.); marco.distanislao@studio.unibo.it (M.D.S.); pierandrea.deiaco@unibo.it (P.D.I.); 2Centro di Studio e Ricerca delle Neoplasie Ginecologiche (CSR), University of Bologna, 40138 Bologna, Italy; stefano.miglietta2@unibo.it (S.M.); giulia.girolimetti3@unibo.it (G.G.); giuseppe.gasparre3@unibo.it (G.G.); annamaria.porcelli@unibo.it (A.M.P.); claudio.zamagni@aosp.bo.it (C.Z.); alessio.morganti2@unibo.it (A.G.M.); 3Department of Medical and Surgical Sciences (DIMEC), University of Bologna, 40138 Bologna, Italy; 4Department of Pharmacy and Biotechnology, University of Bologna, 40126 Bologna, Italy; 5Center for Applied Biomedical Research, Alma Mater Studiorum-University of Bologna, 40138 Bologna, Italy; 6Institute of Hematology, Department of Experimental, Diagnostic and Specialty Medicine, University of Bologna, 40138 Bologna, Italy; lisa.argnani@unibo.it; 7Radiation Oncology, IRCCS Azienda Ospedaliero-Universitaria di Bologna, 40138 Bologna, Italy; martina.ferioli4@unibo.it; 8Department of Experimental, Diagnostic and Specialty Medicine, University of Bologna, 40138 Bologna, Italy; 9Interdepartmental Center for Industrial Research Life Sciences and Technologies for Health, Alma Mater Studiorum-University of Bologna, 40064 Ozzano dell’Emilia, Italy; 10Scientific & Medical Department IGEA S.p.A., 41012 Carpi, Italy; f.deterlizzi@igeamedical.com; 11Oncologia Medica Addarii, IRCCS Azienda Ospedaliero-Universitaria di Bologna, Via Albertoni 15, 40138 Bologna, Italy

**Keywords:** vulvar cancer, electrochemotherapy, cisplatin, bleomycin

## Abstract

**Simple Summary:**

Electrochemotherapy (ECT) is an emerging treatment for solid tumors and an attracting research field due to its clinical results. ECT in association with bleomycin is an effective and safe treatment option in the vulvar cancer palliative setting. With regard to cisplatin (CSP)-based ECT, considering the clear evidence on its efficacy in gynecological tumors, the possibility to improve local control with CSP-based ECT is intriguing and a well-designed randomized clinical trial should be addressed to this issue.

**Abstract:**

Electrochemotherapy (ECT) is an emerging treatment for solid tumors and an attractive research field due to its clinical results. This therapy represents an alternative local treatment to the standard ones and is based on the tumor-directed delivery of non-ablative electrical pulses to maximize the action of specific cytotoxic drugs such as cisplatin (CSP) and bleomycin (BLM) and to promote cancer cell death. Nowadays, ECT is mainly recommended as palliative treatment. However, it can be applied to a wide range of superficial cancers, having an impact in preventing or delaying tumor progression and therefore in improving quality of life. In addition, during the natural history of the tumor, early ECT may improve patient outcomes. Our group has extensive clinical and research experience on ECT in vulvar tumors in the palliative setting, with 70% overall response rate. So far, in most studies, ECT was based on BLM. However, the potential of CSP in this setting seems interesting due to some theoretical advantages. The purpose of this report is to: (i) compare the efficacy of CSP and BLM-based ECT through a systematic literature review; (ii) report the results of our studies on CSP-resistant squamous cell tumors cell lines and the possibility to overcome chemoresistance using ECT; (iii) discuss the future ECT role in gynecological tumors and in particular in vulvar carcinoma.

## 1. Introduction

Vulvar carcinoma (VC) is a rare disease (5% of all gynecological neoplasms) with the highest incidence in the elderly, although, in recent years, the spread of human papilloma virus (HPV) infection in young women changed this trend [1]. Squamous cell carcinoma (SCC) is the most frequent histological type (90% of cases), although several rare entities such as melanoma, extra-mammary Paget’s disease, Bartholin’s gland adenocarcinoma, verrucous carcinoma, basal cell carcinoma (BCC), and sarcoma can occur [2,3,4]. Surgery is the standard treatment, frequently combined with neoadjuvant chemoradiation; alternatively, the treatment is based on exclusive chemoradiation, based on disease site and patient’s performance status [5]. VC is often multifocal and recurs in about one third of cases with 70% 5-year overall survival rate [5]. When VC recurs, the therapeutic options are limited also due to the frequently advanced patient’s age leading to the frequent referral to palliative care alone. Chemotherapy and radiotherapy have been used in VC symptoms palliation. However, tumor response rates are dismal and data on symptoms relief are almost completely lacking.

Furthermore, radiotherapy has obvious limitations being associated with a non-negligible incidence and severity of adverse events (AEs), leading to a worse quality of life (QoL). On the contrary, particularly in the palliative setting, treatments effective in terms of local control but without relevant AEs are ideally required. Unfortunately, even surgery and chemotherapy often do not meet this need. In this scenario, some research groups have proposed electrochemotherapy (ECT) to achieve local tumor control and symptoms relief based on the positive results recorded in skin tumors [6,7,8].

In fact, ECT represents a new therapeutic option of metastatic skin tumors not amenable to surgery or radiotherapy [9]. ECT is based on reversible electroporation (EP) induced by short electrical impulses which increase the permeability of cancer cells membranes to cytotoxic agents. The effect of EP allows drugs with low permeability to concentrate inside the cells, strongly improving their therapeutic efficacy.

In this paper, we summarized the principles of ECT, the mechanism of action of bleomycin (BLM) and cisplatin (CSP), the most frequently drugs combined with EP, and evidence on ECT in VC. Furthermore, we performed a systematic review on CSP-based ECT, reported on the possibility to overcome CSP resistance using EP in SCC cells, and discuss the need and feasibility of a clinical trial aimed to address most unanswered questions on ECT in VC.

## 2. Principles of Electrochemotherapy and the New Palliative Care in Vulvar Cancer

EP is induced by a train of square wave electrical impulses, with the potential development of pores and their equally rapid disappearance (from few seconds to several minutes) after exposure to the electric field. However, being small and short-lived, the hypothetical pore generation caused by EP has not yet been observed directly, due to the microscale dimension and fragile structure of pores and cells [10,11]. For these reasons, even if evidence seems to suggest that, the pore formation on cell membranes is not completely understood [12]. In general terms, it is accepted that EP promotes structural changes in the cell membrane lipid bilayer which allows higher permeability [13]. This transient condition may be exploited to promote the entry into the cells of non-permeable drugs administered intravenously (IV) or intratumorally [14]. The basic ECT principles are as follows: (1) the peak concentration of chemotherapy in the tumor occurs at the same time of the electrical impulses; (2) the whole tumor mass must be covered by the electric field; (3) to this end, specific electrode sets must be chosen, in terms of depth, size, and shape (plate and needle) [15]. Electrodes may be used as needles arranged in parallel rows, with a space of 4 mm between them (recommended for small nodules), or as needles arranged in a hexagonal manner, ideal for nodules > 1 cm [10,15]. Needles can reach up to 3 cm depth.

Several drugs have been tested in combination with EP: daunorubicin, doxorubicin, etoposide, paclitaxel, actinomycin D, adriamicin, mitomycin C, 5-fluorouracil, vinblastine, vincristine, gemcitabine, cyclophosphamide, and carboplatin [15]. Currently, the two most frequently used drugs are BLM and CSP. BLM is administered IV (15,000 IU/m^2^ as a bolus injected in 30–60 s) or intratumorally at the following concentrations: 1000 IU/cm^3^ for small lesions and 250 IU/cm^3^ for large lesions [16]. Similarly, also the dose of CSP depends on tumor size, ranging between 0.5 to 2 mg/cm^3^ [16].

In the above description, we considered only the equipment described in the European Standard Operating Procedures on Electrochemotherapy (ESOPE) procedures although other types of electroporators, needles, drugs and dosages, were described in the literature [17,18,19,20,21].

BLM and CSP have low membrane permeability but after EP, their intracellular concentration increases up to 1000 and 80 fold, respectively (Figure 1) [14].

BLM is an antibiotic with anticancer activity which exerts its effect on DNA through a multistep process, although the mechanism of action is unknown in detail. Some data suggest that BLM primarily acts through bonding with metal ions and then formation of metallobleomycin complexes. The latter promotes the generation of reactive oxygen species leading to failure of DNA function by inhibiting DNA, RNA, and protein synthesis [22]. In addition, CSP anti-cancer activity is based on DNA lesions’ generation, produced by two different mechanisms: (1) CSP-DNA cross-links inhibit DNA replication and induce cell apoptosis; (2) CSP adducts block RNA elongation and therefore gene transcription, contributing to cell death [23,24]. Figure 2 summarizes the main mechanisms involved in ECT response.

At least two other phenomena are involved in enhancing the final ECT antitumor effect: improved immune response and “vascular lock” (Figure 2) [25,26]. With regard to immune response, it has been suggested that the death of electroporated cells could release tumor antigens recognized by the dendritic cells that trigger an adaptive immune anti-cancer response [26,27,28,29]. In addition, EP increases the diameter of blood vessels within the tumor and produces constriction of arterioles ending up in the “vascular blockage”, which enhances drug trapping and accumulation within the tumor [30]. These mechanisms harm the actively dividing cancer cells more than the non-dividing cells of the surrounding healthy tissue.

In 1993, the first ECT treated tumors, metastatic skin lesions of the head and neck, were described [31]. Over almost 30 years of active research, ECT has been tested and applied in skin metastases from melanoma, breast, and head and neck tumors, and in primary breast, pancreas, colon, and VC [9,32,33,34]. ECT is currently used as an alternative palliative treatment option after standard therapies to improve patients’ QoL. However, the National Institute for Health and Clinical Excellence (NICE) suggests ECT among the possible therapies for: (i) primary BCC and SCC inaccessible or difficult to treat with standard treatments and (ii) skin metastases from non-cutaneous cancers and melanoma [35]. In addition, the German Arbeitsgemeinschaft Dermatologische Onkologie guidelines consider ECT as an alternative treatment of loco-regional relapses from melanoma [36]. Moreover, ECT has been recognized by some national health services for the treatment of skin metastases from various tumors with different histologic types [34]. In 2006, the procedures for ECT have been drawn up by a group of experts, greatly facilitating the introduction and diffusion of this treatment and in 2018, a group of pan-European experts proposed a consensus opinion to update the standard operating procedures based on the gained experience [37,38].

Based on the ECT efficacy in the palliative treatment of skin metastasis, our group published in 2013 the first results on ECT in VC reporting with high local control rates (80%) and significant symptoms improvement [6]. Subsequently, other groups confirmed our results [7,39]. The most frequently treated histologic type was SCC, but data on few cases of melanoma and vulvar Paget’s disease were also promising. In 2019, we reported the preliminary results of the ELECHTRA study, currently the largest study on BLM-based ECT in VC patients unsuitable for standard therapies. The study confirmed the positive results in terms of local control (about 80%) and therefore we proposed ECT as a viable option in the palliative setting of VC [32]. Based on the available data on 110 patients, the overall response rate (ORR) ranges between 60% and 83% in vulvar SCC (Table 1). However, data on QoL were not reported in these studies, while the analyses of this endpoint from the ELECHTRA study have been recently published [40]. Moreover, we treated recurrent and metastatic vaginal tumors with promising results (ORR: 67%) [41] and we tried to avoid extended surgery in VC using ECT as neoadjuvant therapy. Nevertheless, the results of the last two reports were based on small cohort of patients and therefore, they can be considered just as hypothesis-generating pilot studies. Papers reporting on BLM-based ECT in VC patients are summarized in Table 1. It should be emphasized that only BLM-based ECT was tested in gynecological tumors (GT), although CSP represents the most effective drugs in these cancers.

## 3. Systematic Review of Studies on Electroporation Combined with Cisplatin

We performed two systematic reviews. The first aimed to evaluate data available from clinical studies, whereas the second one was focused on in vitro and in vivo studies carried out on GT cell models.

The two systematic reviews were conducted in accordance with the PRISMA Statement principles [44]. PubMed, Web of Knowledge, and Cochrane Library databases were systematically searched for original articles analyzing the efficacy of CSP-based ECT in different cancers (last updated search 1 March 2021). Relevant studies were selected using the Boolean combination of the following key terms: “electrochemotherapy OR ECT” AND “cancer OR tumor OR tumour OR neoplasia OR tumors OR tumours OR cancers” AND “cisplatin OR platin”. Additionally, the reference list of reviews, meta-analyses, and all original studies were hand-searched to acquire further relevant studies missed from the initial electronic search.

Eligible studies were required to meet the following inclusion criteria: (i) investigations on CSP-based ECT; (ii) evaluation of the effectiveness of CSP-ECT in human cancer. Exclusion criteria for the first systematic review were: (i) meta-analyses, reviews, and editorials; (ii) non-human cancers; (iii) in vitro studies; (iv) non-English articles. When more than one study had been published sharing part of the same patients’ population, only the most recent and complete study was selected. For the second review, based on the above described search strategy, the in vitro and in vivo studies were further analyzed in order to select the ones on GT cell lines (Figure 3).

After removing duplicated studies, two investigators (GR and AMP) independently checked titles and abstracts of the retrieved articles and judged their eligibility. Then, the entire text of potentially eligible studies was evaluated to assess appropriateness of inclusion in this systematic review. The same two authors independently extracted the following data from the selected papers: (1) first author, publication year, and aim; (2) cancer type, cancer type, sample size; (3) CSP administration route and dose; (4) time between CSP administration and EP; (5) clinical outcome.

From the initial search, we found a total of 138 studies, two of which were duplicated; 125 were excluded based on the inclusion criteria, while 10 were judged adequate for analysis [38,45,46,47,48,49,50,51,52,53] (Figure 3). The results are summarized in Table 2.

Treated tumors were primary and metastatic skin neoplasms including melanoma, basal cell carcinoma, breast cancer skin metastases, and others. No GTs were included in any of the selected studies. ECT was performed after failure of the conventional treatments as exclusive therapy (seven studies) or in combination with other therapeutic procedures (three studies). CSP was administered intratumorally (1–2 min before EP) except for one study where CSP was administered three hours before EP in order to provide optimal drug concentration in the tumor nodules [46]. In particular, Sersa et al. tested EP in combination with concurrent CSP-based chemoimmunotherapy of the tumor nodules [45], Snoj et al. reported on neoadjuvant CSP-based ECT in one patient with anorectal melanoma [49], while Hribernik et al. evaluated efficacy and safety of EP plus CSP after previous treatment with IFN-α [48]. An intratumoral CSP route of administration was preferred in 9 out of 10 studies. In the Sersa et al. study, the time between CSP injection and EP was at least 3 h versus 1–2 min of the intratumoral route.

Most studies were retrospective and included a small number of patients. No randomized controlled trials were retrieved. Overall, the selected studies reported on 423 lesions treated with EP plus CSP, the setting was mainly palliative, and tumor response rates ranged between 40% and 100%. Retreatment was described in 9 studies (Table 2) and was performed in cases without CR after the first ECT with only one study reporting on repeated ECT after PD [38]. In all retreated cases, a clinical response was recorded, except for only two cases [53]. Regarding the occurrence of PD after repeated ECT, the authors observed that this was a rare occurrence and they concluded that, given the evidence from other studies, their findings should not discourage the combination of CSP and EP.

Generally, response to therapy was assessed 4 weeks after ECT using the Response Evaluation Criteria in Solid Tumors (RECIST). No serious AEs such as edema, ulceration, and depigmentation occurred in the analyzed studies.

Regarding the in vitro and in vivo studies, 36 reports were identified: 33 were excluded because not including GT cell lines while only three studies evaluating EP in GT cell models were selected for the analysis. Effectiveness of EP and CSP in GTs has been so far described by a few reports.

The first evidence dates back to 1998, when Cemazar et al. reported on EP plus CSP in ovarian cancer cells [54]. The aim of this in vitro study was to evaluate the role of EP in promoting CSP efficacy in the characterized human ovarian carcinoma IGROV1cells and their resistant subclone IGROV1/DDP. Both cell lines were treated with CSP alone or in association with EP for an exposure time of 5 min. The results showed that CSP cytotoxicity was significantly potentiated by EP in both cell lines although the IGROV1/DDP subclone exhibited a 50-fold higher resistance. These preliminary results proved that EP enhances CSP activity and cytotoxicity on IGROV1 and IGROV1/DDP cells and paved the way for further investigations.

A subsequent study investigated the cytotoxic effect of 5-fluorouracil and CSP on two human ovarian cancer cell lines, OvBH-1 and SKOV-3, and EP ability to reduce the effective drug dose [55]. The authors reported that EP with CSP showed a major enhancement in drug transport and in cytotoxic activity in both cell lines. This result is very interesting because both cell lines were CSP-resistant, which, as mentioned, is a largely common circumstance in ovarian cancer chemotherapy. In fact, resistance is a main issue in CSP-based chemotherapy and identification of novel drug delivery systems remains an unmet clinical need.

Recently, Łapinska et al. further analyzed the role of CSP-based ECT in ovarian cancer in combination with estradiol [56]. Indeed, it is well known that the biological mechanisms of ovarian cancer development and progression are tightly linked to estrogen hormones. Estrogens exert a direct action on ovarian cells that may promote tumor development, e.g., producing DNA damage due to enhanced proliferation; 17β-estradiol is usually over-expressed in ovarian cancer patients, but its role is not completely clear. The authors aimed to evaluate 17β-estradiol effect in combination with ECT and CSP in ovarian cancer cells. The MDAH-2774 cell line was used. EP was first evaluated alone (using 3 different protocols) and then by adding CSP with or without a preincubation with 17β-estradiol. EP alone achieved a strong cytotoxic effect with 50% reduction in cell viability while CSP addition enhanced this effect. Electroporated cells, irrespective of the protocol, when pre-incubated with 17β-estradiol, showed reduced mitochondrial activity, which was used as a surrogate of cell viability. At the same time, the authors highlighted that estradiol reduced EP effectiveness, suggesting the existence of other mechanisms not related to changes in the cell membrane permeability.

Our review shows the lack of in vivo studies and that the in vitro analyses have been so far performed only on ovarian cancer cells; we did not find studies on squamous cell carcinoma cells which is the most frequent GT histologic types. However, all reported in vitro studies have shown an enhanced EP therapeutic effect by combination with CSP, confirming the potential effectiveness of this drug in the ECT setting.

## 4. Electroporation Increased the Sensitivity to Cisplatin of Uterine Cervical Chemoresistant Cancer Cells

In order to test if EP is able to overcome platinum resistance in SCC human cervical cancer, CSP-sensitive 2008 cell line and its syngeneic CSP-resistant C13 cell line (generated by growth in 1 mM CSP for 13 months) were used. Both cell lines were grown in RPMI 1640 medium, supplemented with 10% fetal bovine serum, 2 mmol/L L-glutamine, 100 units/mL penicillin, and 100 mg/mL streptomycin in a humidified incubator at 37 °C with 5% CO_2_.

When 2008 and C13 cancer cells reached 70% confluence, they were harvested with 0.5% trypsin, counted, and resuspended in medium (1 × 10^6^ cells/mL). A volume of 200 μL of cell suspension was placed in an EP cuvette with 0.2 cm gap between the electrodes (Sigma-Aldrich, St. Louis, MO, USA). CSP (Accord Healthcare) was added to 2008 and C13 cell suspensions at different concentrations (0.5, 1, 2, 4, 8, 16 μM). Cuvettes with CSP-treated cells were placed in the Cliniporator ^®^ (IGEA S.p.A) and ECT was performed following pre-set parameters (400 V, 8 pulses, 100 μs). The EP efficiency was evaluated by counting cells stained with trypan blue dye and then cell suspensions were incubated at 37 °C and 5% CO_2_ for 30 min. Subsequently, cell suspensions were diluted (1:10) and seeded in 24-wells plates. In parallel, non-electroporated cells were seeded as negative control. Furthermore, the basal effect of CSP in chemoresistant and chemosensitive cell lines was evaluated by measuring cell viability after 72 h of treatment with different drug concentrations.

Cell viability was assessed by using sulforhodamine B (SRB), an anionic dye which forms electrostatic complexes with basic protein residues in acid conditions. At the end of each treatment time, cells were fixed with 50% trichloroacetic acid for 1 h at 4 °C, washed 5 times with water, and dried at room temperature. Attached cells were stained for 30 min with 0.4% SRB diluted in 1% acetic acid at room temperature. Subsequently, cells were washed 4 times with 1% acetic acid in order to remove the excess of dye. The dye bound to the cells was finally solubilized with 10 mM Tris (pH 10.5) and the relative absorbance of SRB was detected using a plate reader (VICTOR3 1420 Multilabel Counter-PerkinElmer, Turku, Finland) at the wavelength of 560 nm.

In order to evaluate the in vitro effect of EP combined with cisplatin, we used CSP-sensitive 2008 cervical cancer cell line and its syngeneic chemoresistant C13 counterpart. The effect of different CSP concentrations was determined in terms of cell viability (Figure 4A), confirming a higher CSP-related half-maximal effective concentration (EC50) in the chemoresistant cell line compared to the sensitive counterpart (Figure 4B). Moreover, we ruled out a possible effect of EP alone, at the conditions we used, on cell viability unrelated to cisplatin, on both cell lines (Figure 4C). We next proceeded to test if the CSP cytotoxic effects could be boosted through EP, and to this aim, we tested increasing drug concentrations: the viability of both cell lines was significantly reduced by ECT, in a CSP dose-dependent manner (Figure 4D). Interestingly, the effect of ECT was more prominent in the CSP-chemoresistant cell line.

## 5. Discussion

The topic of our review is ECT, a relatively new anticancer treatment, primarily used as palliative therapy of skin metastases from different tumors. The progressive spread of ECT has led to a gradual widening of the indications for this treatment. Our group tested palliative ECT in GTs and particularly in VC, with 80% ORR rates. Although the most widely used drug in combination with EP is BLM, in our systematic review, we found 10 studies on skin metastases treated with intratumoral or intravenous CSP administered with EP, with the same indications and results of BLM-based ECT. Moreover, focusing on GT only, we reviewed the literature on the in vitro and in vivo studies about CSP-based ECT, finding only three in vitro studies. All of the latter concerned ovarian cancer cells and showed promising results with this combined modality treatment. Finally, we reported the results of our in vitro study, representing the first analysis on CSP-chemoresistant squamous cells and showing that EP is able to "sensitize" cells to CSP. These results could open up new therapeutic scenarios in the treatment of VC, largely represented by squamous cell carcinomas.

VC affects old women, a heterogeneous group of patients due to several comorbidities and organ dysfunction, leading to a high demand for resources in terms of social support. Palliation of advanced and relapsing VC represents an important health and social issue. However, effective palliative treatments of VC are lacking, being the local control rates after chemotherapy relatively low and due to similar results and high toxicity rates recorded in radiotherapy series (Appendix A). Moreover, in the available studies on VC chemotherapy and/or radiotherapy, the results in terms of QoL were generally not reported.

More specifically, the response rates achievable with different chemotherapy regimens are very low, particularly in terms of complete response with percentages ranging from 0.0% to 9.7% [57,58,59]. Similarly, radiotherapy results are poor, with ORRs ranging among 31.0% and 60.0%, with complete response rate being 20% only, and with all patients having bulky tumors dying within 5 years [60,61,62]. Furthermore, radiotherapy does not appear to be an ideal treatment in the palliative setting, being associated with a non-negligible incidence and severity of AEs, able to worsening patients’ quality of life. In fact, in terms of acute toxicity, the following AEs are common: mucocutaneous vulvar, perineal, and inguinal folds inflammation and potentially hematological toxicity. Moreover, in terms of late toxicity, the following AEs have been reported: teleangectasia and atrophy of skin and vulvar mucosa, dryness of vulvar and vaginal mucosa, and avascular necrosis of the femoral head [63].

In contrast, as shown by our literature review, studies on BLM-based ECT in patients with recurrent VC showed promising results in terms of symptoms relief, local control (60–80%), and toxicity profile. Based on this evidence, the question could be: compared to BLM-based ECT, may CSP-based ECT further improve local disease control without increasing AEs rates and severity?

So far, BLM was the drug most frequently combined with ECT. However, CSP offers some theoretical advantages that should be considered. In particular, CSP may be preferred over BLM in patients with renal disease or in older patients, typically affected by impaired or at least reduced renal function. BLM is associated with higher kidney, skin, and lung toxicity and these aspects should be considered when a single patient requires multiple treatments. Moreover, the time from drug injection to EP is relatively longer with BLM (from 5 to 8 min), whereas intratumorally injected CSP requires a shorter time. Regarding CSP-based ECT toxicity, considering the few studies on EP + CSP, available data on AEs are mainly related to standard chemotherapy. However, toxicity should be not relevant considering that CSP is administered only intratumorally according to the ESOPE guidelines.

The present systematic review on CSP-based ECT has several limitations. In fact, the selected studies have the following characteristics: small sample size, different treated neoplasms, retrospective design, only intralesional administration (except one), and an overall high risk of bias. However, except for one study, the authors homogeneously reported more than satisfactory results. The comparison with BLM-based ECT, particularly in terms of AEs, is also difficult considering the different administration routes, almost exclusively intravenous for BLM and intratumoral for CSP. Platinum-based compounds represent the most effective chemotherapy drugs in epithelial GTs. Literature data show promising results with EP combined to intratumoral administration of CSP in melanoma and other skin metastasis, but the ESOPE guidelines recommend this treatment only for lesions smaller than 3 cm [37]. In our previous study, we included also lesions ≥ 3 cm, where the intralesion route in not pursuible for all VC recurrence, and IV administration seems the best solution [32]. However, the commonly used CSP may not be easy to handle in elderly patients due to the large fluid load required to prevent renal toxicity. The latter, due to tubular damage both in the acute and chronic forms, and the risk of peripheral neuropathy represent the main factors limiting the feasibility of CSP-based chemotherapy in the older population. Studies on ovarian cancer show that, compared to CSP, carboplatin has comparable efficacy with lower renal, gastrointestinal, and hematological AEs rates, with shorter duration of peripheral neuropathy. Based on this evidence, carboplatin plus paclitaxel is the preferred regimen in patients with ovarian cancer and >75 years old.

Regarding CSP combined to EP in GT cell models, data are very limited. Our literature review led to the identification of only three studies. However, all papers reported an improvement of CSP efficacy when combined with EP, especially in CSP resistant models. On the same wavelength, our in vitro study showed that EP is able to improve CSP effectiveness in a CSP dose-dependent manner and particularly in CSP resistant cells. Taken togheter, all these results suggest that EP could represent an efficient escamotage to bypass platinum resistance; morever, this evidence may pave the way to the translation of CSP coupled EP treatement in ovarian cancer patients who develop resistance to platinum based chemotherapy, however difficult to apply in clinical practice.

Interestingly, although VC is the most common cancer treated with EP, no in vitro evaluations are reported in literature. Actually, all clinical studies in VC were directly performed in a palliative human setting and without preclinical data. Nevertheless, the recorded outcomes were excellent in most patients.

Platinum compounds are generally effective against SCC cells, but the development of chemoresistance is common. This scenario lead to the need of novel drug delivery systems to improve the efficacy of these drugs. Our in vitro study showed the possibility to improve CSP efficacy, by combination with EP, based on the viability of two cell lines, in a CSP dose-dependent manner and with a more evident effect of combined modality treatment in CSP-resistant cells. Translated into a hypothetical clinical practice, this setting could mimic VC recurrences after developement of chemoresistance where EP could be able to restore chemosensitivity.

## 6. Conclusions

Based on our and other studies, ECT with BLM is an effective and safe treatment option in the VC palliative setting. Considering the clear evidence on CSP efficacy in GTs, the chance to improve local control with CSP-based ECT is intriguing; a well designed randomized clinical trial including both drugs should be addressed to this issue, although CSP in clinical practice is more difficult to manage compared to BLM.

## Figures and Tables

**Figure 1 cancers-13-01993-f001:**
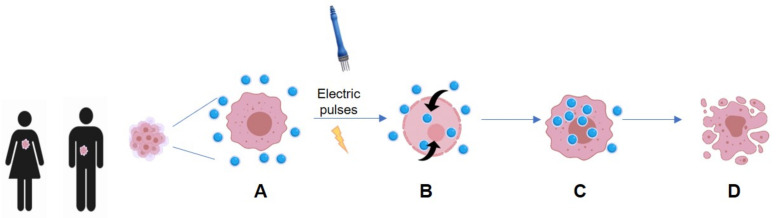
(**A**) After administration, the drug reaches the tumor cells accumulating outside them, being their entry hindered by the cytoplasmic membrane. (**B**) Applications of EP promote structural changes in the cell membrane lipid bilayer which allows higher permeability. (**C**) The drug can enter the cell. (**D**) The drug can exert its action promoting irreversible damage which causes cell death.

**Figure 2 cancers-13-01993-f002:**
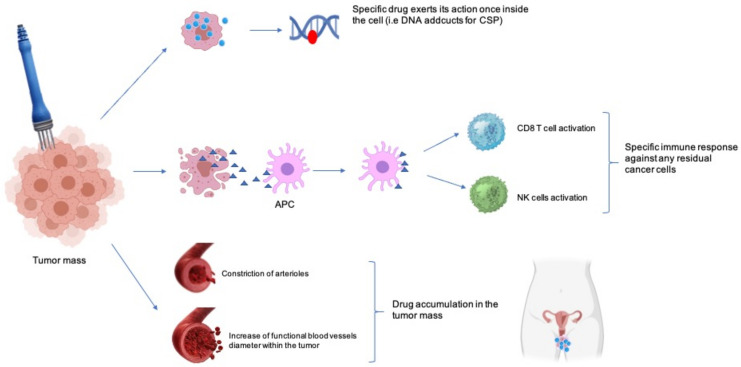
Main mechanisms involved in ECT response: (1) higher drug uptake (higher panel), (2) immunogenic response (middle panel), and (3) vascular lock (lower panel); CSP: cisplatin; APC: antigen presenting cells; NK: natural killer.

**Figure 3 cancers-13-01993-f003:**
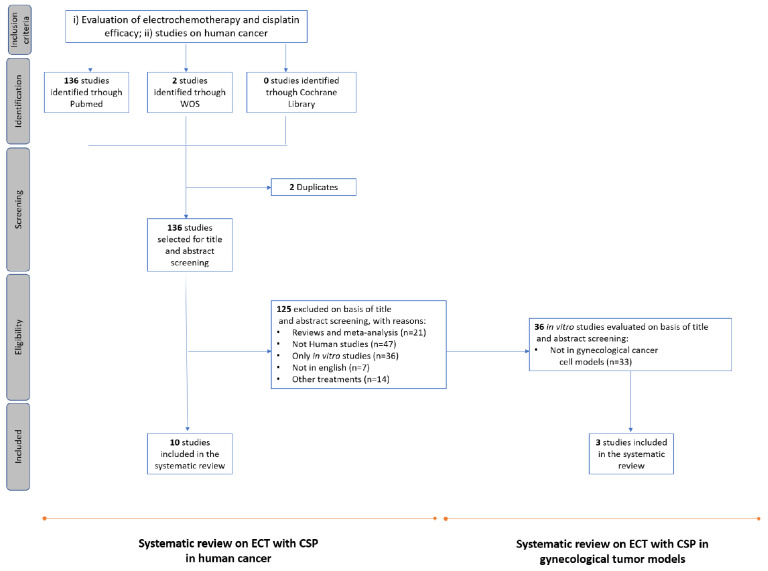
Flowchart of literature search and selection process of eligible studies.

**Figure 4 cancers-13-01993-f004:**
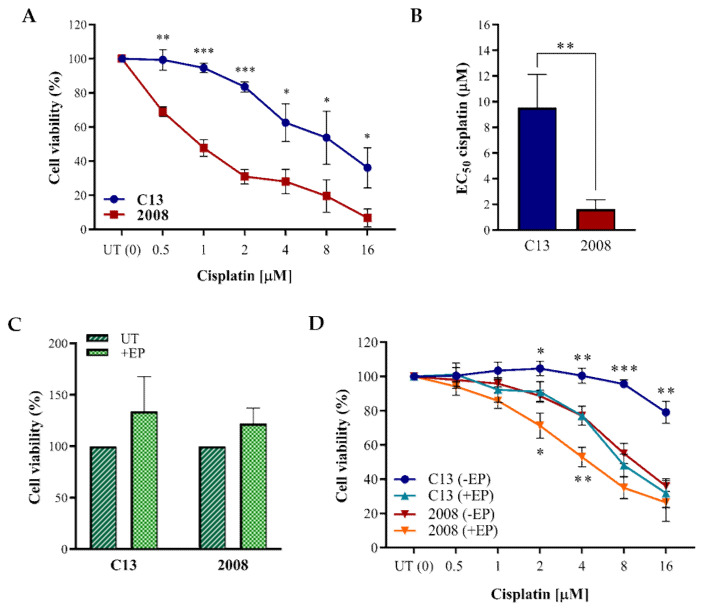
(**A**) Cell viability of C13 and 2008 cancer cell lines after 72 h of treatment with different CSP concentrations. Data are mean ± SD (n = 3). (**B**) CSP EC50 in C13 and 2008 cell lines calculated after 72 h of treatment. EC50 indicates the CSP concentration that results in a 50% decrease of cells viability. Data are mean ± SD (n = 3). (**C**) Cell viability of C13 and 2008 cancer cell lines after 72 h from the EP performed at 400 V, with 8 pulses in 100 µs. Data (mean ± SD; n = 3) were normalized to the cell viability of non-electroporated cells (UT) reported as 100%. C13 UT vs EP C13: *p* > 0.05; 2008 UT vs EP 2008: *p* > 0.05. (**D**) Cell viability of C13 and 2008 cancer cell lines after 72 h from EP treatment, using increasing CSP concentration in combination or not with EP (±EP). Data (mean ± SD; n = 3) were normalized on the cell viability of CSP-UT. Asterisks indicate the significance of data related to EP treated vs. untreated cells. All *p*-values were obtained with Student’s *t*-test by using GraphPad Prism (* *p* ≤ 0.05; ** *p* ≤ 0.01; *** *p* ≤ 0.001). UT: untreated cells (non-electroporated cells).

**Table 1 cancers-13-01993-t001:** Studies on bleomycin-based electrochemotherapy in vulvar cancer.

Author and Year; Aim	Number of VC Patients	Histology	BLM Administration		Time between BLM Administration and EP	Electric Field	Electrodes	Cliniporator	Evaluation
			Route	Dose					
Perrone et al., 2013 [6] Aim: To evaluate safety, local response, feasibility, and QoL of EP + BLM.	10	9 SCC (1 pt lost at FU)	i.v. injected as bolus (30 s)	15,000 UI/m^2^	EP started 8 min after bolus and treatment was completed 28 min after infusion end	Pulse delivery frequency: 5 kHz at a duration of 100 μs.	Type III electrodes placed into the lesion Electrodes gently inserted into the skin of the area affected at a depth of one centimeter; procedure repeated to cover the entire area to be treated.	IGEA Italy	After at least 4 weeks CR: 62.5% PR: 12.5% NC: 12.5% PD: 12.5%
Perrone et al., 2015 [42] Aim: To assess tumor response, symptoms relief, and local tumor control after EP + BLM	25	25 SCC	Same of ref [6]		Same of ref [6]	Same of ref [6]	Same of ref [6]	IGEA Italy	OR: 80% CR: 52% PR: 28% SD: 12% PD: 8% After 4 weeks: local tumor control 91% After 6-months: local tumor control 53%, After 1 year: OS 34%
Pellegrino et al., 2016 [39] Aim: To evaluate safety, local tumor response, and symptoms relief after ECT	10	9 SCC, 1 Paget’s	i.v, in a bolus lasting 60–90 s.		Same of ref [6]	Pulse delivery frequency: 5 kHz at a duration of 100 μs	Electrodes gently inserted into the skin of the area affected at a depth of one centimeter; procedure repeated to cover the entire area to be treated. EP delivered by two types of needle electrodes (parallel arrays finger and hexagonal arrays N-10-HG-10 mm, suitable for bigger tumors)	IGEA Italy	OR: 60% CR: 20% PR: 40% SD: 20% PD: 20%
Perrone et al., 2018 [43] Aim: To test neo-adjuvant ECT	9	9 SCC	Same of ref [6]		Same of ref [6]	Same of ref [6]	Same of ref [6]	IGEA Italy	CR: 11.1% PR: 66.7% SD: 22.2%.
Perrone et al., 2019 [32] Aim: To investigate local control after ECT	61	57 SCC 3 Paget’s 1 melanoma Tumor response evaluated in 55 pts	Same of ref [6]		Same of ref [6]	Same of ref [6]	Same of ref [6]	IGEA Italy	After at least 2 weeks OR: 83.6% CR: 52.7% PR: 30.9% SD: 10.9% PD: 1.8%
Corrado et al., 2020 [7] Aim: To evaluate clinical outcome and side effects after ECT	15	14 SCC 1 CS	Same of ref [6]		Same of ref [6]	Same of ref [6]	Same of ref [6]	IGEA Italy	After at least 4 weeks CR: 13.3% PR: 66.6% SD: 13.3% PD: 6.6%

Legend: CR: complete response; CS: carcinosarcoma; i.v.: intravenous; PR: partial response; PD: progressive disease; OR: objective response; QoL: quality of life; SCC: squamous cell carcinoma; VC: vulvar cancer.

**Table 2 cancers-13-01993-t002:** Studies on cisplatin-based electrochemotherapy.

Author and Year; Aim	Type of Cancer	Number of pts and Lesions	Type of Sedation	CST Administration		Time between CSP Administration and EP	Electric Field	Electrodes	Cliniporator	Evaluation	Retreatment with ECT	Toxicity
				Route	Dose							
Sersa et al., 1998 [45] Aim: To assess response after EP + CSP.	Malignant melanoma, SCC, BCC.	4 pts, 30 nodules. 19 nodules treated with EP and CSP, 1 with EP, 5 with intratumor CSP, 5 no treatment	NR	i.t.	0.25–2 mg depending on the size of nodule Concentration of 2 mg/mL of saline solution	1–2 min	Square wave EO of 100 µs, 910 V amplitude (amplitude to electrode distance ratio 1300 V/cm), frequency 1 Hz.:	Two parallel stainless-steel electrodes (thickness 1 mm; width 7 mm; length 14 mm, with rounded tips and inner distance between them 7 mm)	Jouan GHT 1287 (Jouan, France).	After at least 4 weeks EP + CSP: CR: 100% CSP alone: CR: 40%, PD: 60%.	In cases without CR	Mild; Local
Sersa et al., 2000 [47] Aim: To evaluate antitumor efficacy of EP + CSP	Melanoma	10 pts, 133 nodules. 82 nodules treated with EP and CSP, 27 treated with CSP, 2 treated with EP, 22 untreated.	NR	i.t.	Same of ref [45]	1–2 min.	Same of ref [45]	Same of ref [45]	Same of ref [45]	After at least 4 weeks EP + CSP: OR: 78%; CR: 68%; PR: 10%; PD: 7%; NC: 15%. CSP alone: OR: 38%, CR: 19%; PR: 19%. PD: 33%, NC: 30%. EP alone; PD: 50%; PR: 50% UT nodules PD: 64%; NC: 36%	In case of not CR	Mild; Local
Sersa et al., 2000 [46] Aim: To evaluate the effectiveness EP + CSP	Skin metastases from malignant melanoma	9 pts, 27 lesions vs 18 lesions	NR	i.v.	Vinblastine (4 mg/m^2^), oral lomustine (80 mg/m^2^) on Day 1, intravenous cisplatin (20 mg/m^2^) on Days 2 ± 5, and interferon-2b 3 mg/m^2^ on Days 4–7	At least 3 h after the infusion start. EP performed on day 4 of chemoimmunotherapy protocol and the same day of the CSP administration during the daily chemotherapy	1300 V/cm, 8 pulses, 99 µs, 1 Hz			After at least 4 weeks, EP + CSP: OR: 48% CR: 11% PR: 37% NC: 40.7% PD: 11.1% Vinblastin alone: OR: 22% CR: 11.1% PR: 11.1% NC: 38.9% PD: 38.9%	In case of PD	Mild; Local
Sersa et al., 2003 [52] Aim: To evaluate efficacy of EP + CSP	Malignant melanoma	14 pts, 211 lesions	NR	i.t.	1 mg/cm^3^	NA	Square wave 100 µs, 910 V amplitude (amplitude to electrode distance ratio 1300 V/cm), frequency 1 Hz;	Two parallel stainless-steel electrodes (thickness 1 mm; width 7 mm; length 14 mm, with rounded tips and inner distance between them 7 mm)	Same of ref [45]	After at least 4 weeks OR: 82% CR: 70.1% PR: 10.9% NC: 11.4% PD: 7.6 %	In cases with no CR	Mild; Local
Rebersek et al., 2004 [51] Aim: To evaluate efficacy of EP + CSP	Metastatic breast cancer Prior standard treatment or refused other standard treatments.	6 pts with 26 cutaneous lesions. 12 lesions treated with EP + CSP and 6 with CSP alone; 8 untreated	LA	i.t.	Same of ref [45]	1–2 min.	Square wave EP locally on cutaneous tumor lesions;	Two superficial plate electrodes (thickness 1 mm, width 7 mm, length 14 mm, inner distance between them 7 mm). EP amplitude 910 V, duration 100 µs, and frequency 1 Hz	GHT 1287 (Jouan, France). In lesions > 7 mm, treatment performed in several EP runs with electrodes repositioning	after at least 4 weeks EP + CSP treated lesions OR: 100% CR: 33%; PR: 67% CSP alone treated group PR: 83% PD: 17% UT nodules PD: 75%; NC: 25%	In cases with no CR	Mild, Local
Snoj et al., 2005 [49] Aim: To evaluate achievement of local sphincter-saving excision	Anorectal malignant melanoma	1 pt, 1 lesion	GA	i.t.	Total dose of 6 mg Concentration of 2 mg/mL of H20	2–5 min	ELECTRIC FIELD: At each application 8 EP 730 V and 100 µs duration delivered at a frequency of 5 kHz between each pair of neighboring electrodes (8 mm apart). Thus, 192 EP delivered at each application			after at least 4 weeks PR (from 6.2 cm^3^ to 3.8 cm^3^)	yes	NR
Marty M, et al., 2006 [38] Aim: To evaluate efficacy and safety of BLM- and CSP-based ECT	Skin and subcutaneous metastases from malignant melanoma carcinoma and sarcoma	44 lesions Part of a larger study including a total of 41 pts and 171 lesions treated with EP + BLM or EP + CSP	GA or LA	i.t.	0.5–2 mg/cm^3^ of tissue	Within 2 min	1300 V/cm, 8 pulses, 100 µs, 1 or 5000 Hz (type I), 1000 V/cm, 8 pulses, 100 µs, 1 or 5000 Hz (type II), 1000 V/cm, 96 pulses, 100 µs, 5000 Hz (type III)	Type I, II and III electrodes		after at least 4 weeks CR: 75.4%	In case of PD	Mild, Local
Campana L et al., 2014 [53] Aim: To evaluate ECT efficacy and safety	SCC	2 pts; Number of lesions not reported Comparison with BLM alone	GA	i.t	0.5–2 mg/cm^3^ of tissue	NA	8 pulses, 100 µs, 5000 Hz. 960V;	Two plate electrodes	IGEA, Italy	after at least 4 weeks SD 50% PD: 50%	Not performed	Mild, Local
Hribernik et al., 2016 [48] Aim: To evaluate ECT after treatment with IFN-α	Cutaneous melanoma	2 pts, 3 lesions Part of a larger study including a total of 5 pts, 111 lesions treated with EP + BLM or EP + CSP	GA or LA	i.t.	Same of ref [45]	Within 2 min	1300 V/cm, 8 pulses, 99 µs, 1 Hz			after at least 4 weeks CR: 100%	In cases with no CR	NR
De Giorgi et al., 2020 [50] Aim: To evaluate EP+CSP efficacy and patients’ tolerability	SCC, BCC, or skin metastases from cancer other than melanoma	8 pts, 18 lesions (3 SCC, 4 BCC, 1 skin metastases from breast cancer)	GA	i.t.	Total dose at least 0.441 mL for each treated lesion. Concentration at least 1 mg/mL	Within few minutes	1000 V/cm and 100 µs the pulse duration; 3.8 pulses for each pulse delivering	Linear lamellar electrode (P-30-8B), finger electrodes (F-10-OR) or linear electrodes (N20-4B) were used	EPS02 model	after at least 8 weeks CR: 50% PR: 50% ORR: 100%	In cases with no CR	Mild, Local

Legend: BCC: basal cell carcinoma; CR: complete response; GA: general anaesthesia; i.t.: intratumor; i.v.: intravenous; LA: local anaesthesia; NR: not reported; OR: objective response: ORR: overall response rate; PR: partial response; PD: progressive disease; SCC: squamous cell carcinoma; UT: untreated.

## Data Availability

The data presented in this study are available on request from the corresponding author.

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
