# Peer review of "Electrochemotherapy in Vulvar Cancer and Cisplatin Combined with Electroporation. Systematic Review and In Vitro Studies"

_cancers, 2021, doi:10.3390/cancers13091993_

Round 1

Reviewer 1 Report

This is an interesting manuscript which gives a review of ECT in general with a focused review on the use of ECT for vulvar cancer. In general the manuscript is good, but there are some aspects that need to be revised.

In the general review of ECT, the authors make some statements that do not necessarily follow the literature. They mention the notion of "pores". While this has been an ongoing debate in the field for several decades, it is generally accepted that actual pores do not form in the membrane. The second aspect is discussed and depicted in Figure 2. While there have been limited reports of an "immune response", there is very little evidence that a specific immune response is generated following ECT. There is only 1 or 2 reports of any patients seeing an effect at a distant site following ECT treatment. This is not sufficient evidence to report a specific immune response is generated.

The authors talk about specific electrode types lines 104-108 as well as rout of administration but leave out other reports and devices as well as other dosing protocols. I would suggest the authors read and possibly add the following papers: Hofmann, et al, IEEE Trans Biomed ENG. 1999, 46(6):752-9; Mir, et al, British J of Cancer, 1998, 77(12):2336-2342; Gilbert, et al, Biochemica, Biophysica Acta, 1997, 1334:9-14; Heller, et al, Cancer, 1998, 83(1):148-157.

In the in vitro experiment reported in the paper, it is curious that in figure 4 there is a difference in the IC50 for cisplatin treated 2008 cells in the absence of EP (comparing Figure 4 B and 4D.

On lines 419 and 421 references are missing.

There are many grammatical and spelling errors

Author Response

REVIEWER 1

This is an interesting manuscript which gives a review of ECT in general with a focused review on the use of ECT for vulvar cancer. In general the manuscript is good, but there are some aspects that need to be revised.

RESPONSE: We thank the reviewer for the comments

In the general review of ECT, the authors make some statements that do not necessarily follow the literature. They mention the notion of "pores". While this has been an ongoing debate in the field for several decades, it is generally accepted that actual pores do not form in the membrane. The second aspect is discussed and depicted in Figure 2. While there have been limited reports of an "immune response", there is very little evidence that a specific immune response is generated following ECT. There is only 1 or 2 reports of any patients seeing an effect at a distant site following ECT treatment. This is not sufficient evidence to report a specific immune response is generated.

RESPONSE:

With regard to the notion of pores, we changed the sentence in order to highlight the lack of a clear mechanism (line 95-101). Reference has been also added.

With regard to the immune response, we changed the sentence to highlight the evidence of this type of response is preliminary. However, we think we should report also this potential mechanism to give to the reader a more complete vision of ECT. Additional references have been added.

The authors talk about specific electrode types lines 104-108 as well as rout of administration but leave out other reports and devices as well as other dosing protocols. I would suggest the authors read and possibly add the following papers: Hofmann, et al, IEEE Trans Biomed ENG. 1999, 46(6):752-9; Mir, et al, British J of Cancer, 1998, 77(12):2336-2342; Gilbert, et al, Biochemica, Biophysica Acta, 1997, 1334:9-14; Heller, et al, Cancer, 1998, 83(1):148-157.

RESPONSE:

Thank you for your reply. However, this is not a review on the use of ECT but on the use of platinum compounds associated with ECT. The general part was meant to be an introduction for those unfamiliar with the theme and in this way had the opportunity to follow it.

In our methodology we refer to specific Standard Operating Procedures of the ESOPE protocol, which has been validated in several studies. This protocol is available only with a single device, the CliniporatorTM with some specific electrodes, plate and row needles, that have been described in the text. Other devices are not delivering the therapy following the ESOPE SOP, published in 2006 and recently updated [Updated standard operating procedures for electrochemotherapy of cutaneous tumours and skin metastases. Gehl J, Sersa G, Matthiessen LW, Muir T, Soden D, Occhini A, Quaglino P, Curatolo P, Campana LG, Kunte C, Clover AJP, Bertino G, Farricha V, Odili J, Dahlstrom K, Benazzo M, Mir LM. Acta Oncol. 2018 Jul;57(7):874-882. doi: 10.1080/0284186X.2018.1454602. Epub 2018 Mar 25. PMID: 29577784] and their efficacy has not yet equally demonstrated.

In the text we added it “In the above description we considered only the ESOPE procedures although other types of electroporators, needles, drugs and dosages are described in the literature”.

The papers have now been cited.

In the in vitro experiment reported in the paper, it is curious that in figure 4 there is a difference in the IC50 for cisplatin treated 2008 cells in the absence of EP (comparing Figure 4 B and 4D.

RESPONSE: We agree with the reviewer that the results are interesting. EC50 for 2008 cells (cisplatin sensitive cells) is indeed lower that EC50 in C13 cells (cisplatin resistant cells). The viability of both cell lines was significantly reduced by EP.

On lines 419 and 421 references are missing

There are many grammatical and spelling errors

RESPONSE: We thank the reviewer for the comments.

Missing references have been added

English has been revised

Reviewer 2 Report

The manuscript is highly interesting, well-written and the topic is relevant. The systemic reviews of the available literature have been conducted properly. The section presenting original research seems scientifically sound.

 I have only minor concerns:

 Tables:

  • Administration route and dose should be divided into two separate columns.
  • Column of clinical outcome: “EVALUATION” should go as a header (. Time between administration and evaluation should go as a different column.
  • Data on ELECTRIC FIELD, ELECTRODE and EVAPORATOR should be separated into three different columns
  • Table 2: Aims of each study should not be listed entirely in one column; the summarized data can be distributed into different column

Discussion:

  • Lines 371-374: repetition of information already given in Introduction section
  • Lines 419 and 421: the numbers of references should be included, instead of writing [REF].

Author Response

REVIEWER 2

The manuscript is highly interesting, well-written and the topic is relevant. The systemic reviews of the available literature have been conducted properly. The section presenting original research seems scientifically sound.

 I have only minor concerns:

 Tables: 

  • Administration route and dose should be divided into two separate columns.
  • Column of clinical outcome: “EVALUATION” should go as a header (. Time between administration and evaluation should go as a different column.
  • Data on ELECTRIC FIELD, ELECTRODE and EVAPORATOR should be separated into three different columns
  • Table 2: Aims of each study should not be listed entirely in one column; the summarized data can be distributed into different column

Discussion:

  • Lines 371-374: repetition of information already given in Introduction section
  • Lines 419 and 421: the numbers of references should be included, instead of writing [REF].

RESPONSE: We thank the reviewer for the comments.

Tables 1 and 2 have been changed according the suggestions. Aim has been shortened

Lines 371-374 have been deleted to avoid repetitions

References have been added

Reviewer 3 Report

In my opinion the manuscript must be completed revised and reformatted. If the authors want to submit a review article, then they can not include experimental data.

Concerning the general description of ECT technique, the authors fail to mention other electric pulses and electrodes that have been adopted for ECT. 

Author Response

REVIEWER 3

In my opinion the manuscript must be completed revised and reformatted. If the authors want to submit a review article, then they can not include experimental data.

  • Concerning the general description of ECT technique, the authors fail to mention other electric pulses and electrodes that have been adopted for ECT. 

RESPONSE: we thank the reviewer for the suggestions. Before to submit the paper, we asked to the editorial team if our manuscript (review with additional research data) was fine to published and they gave us a positive feedback.

RESPONSE:

Thank you for your reply. However, this is not a review on the use of ECT but on the use of platinum compounds associated with ECT. The general part was meant to be an introduction for those unfamiliar with the theme and in this way had the opportunity to follow it.

In our methodology we refer to specific Standard Operating Procedures of the ESOPE protocol, which has been validated in several studies. This protocol is available only with a single device, the CliniporatorTM with some specific electrodes, plate and row needles, that have been described in the text. Other devices are not delivering the therapy following the ESOPE SOP, published in 2006 and recently updated [Updated standard operating procedures for electrochemotherapy of cutaneous tumours and skin metastases. Gehl J, Sersa G, Matthiessen LW, Muir T, Soden D, Occhini A, Quaglino P, Curatolo P, Campana LG, Kunte C, Clover AJP, Bertino G, Farricha V, Odili J, Dahlstrom K, Benazzo M, Mir LM. Acta Oncol. 2018 Jul;57(7):874-882. doi: 10.1080/0284186X.2018.1454602. Epub 2018 Mar 25. PMID: 29577784] and their efficacy has not yet equally demonstrated.

In the text we added it “In the above description we considered only the ESOPE procedures although other types of electroporators, needles, drugs and dosages are described in the literature.

Round 2

Reviewer 3 Report

In my opinion, it is not correct to write a review article and include original data. I would ask the authors to leave out the original data, if they want to submit a review article, or to expand the experimental part, if they want to submit an original article.

Author Response

We thanks the reviewer for the comments.

we think that adding the experimental part can complete and give strength to our observations and hypotheses. Certainly, further experiments will be carried out with different cell lines and will be reported in an original paper. But currently we would prefer to keep the experimental part as it was specified in the abstract and seems to have been appreciated by the other reviewers.